# High proton conductivity through angstrom-porous titania

Yu Ji[1], Guang-Ping Hao[2] ✉, Yong-Tao Tan[3,4], Wenqi Xiong[5,6], Yu Liu[1], Wenzhe Zhou[1], Dai-Ming Tang[7], Renzhi Ma[7], Shengjun Yuan[6], Takayoshi Sasaki[7], Marcelo Lozada-Hidalgo[3,4] ✉, Andre K. Geim[3,4] ✉ & Pengzhan Sun[1] ✉

Two dimensional (2D) crystals have attracted strong interest as a new class of proton-conducting materials that can block atoms, molecules and ions while allowing proton transport through the atomically thin basal planes. Although 2D materials exhibit this perfect selectivity, the reported proton conductivities have been relatively low. Here we show that vacancy-rich titania monolayers are highly permeable to protons while remaining impermeable to helium with proton conductivity exceeding 100 S cm$^{-2}$ at 200 °C and surpassing targets set by industry roadmaps. The fast and selective proton transport is attributed to an extremely high density of titanium-atom vacancies (one per square nm), which effectively turns titania monolayers into angstrom-scale sieves. Our findings highlight the potential of 2D oxides as membrane materials for hydrogen-based technologies.

Proton-permeable two dimensional (2D) crystals, such as graphene and hexagonal boron nitride (hBN), display high transparency to thermal protons while retaining complete impermeability to all ions and gases[1–5]. Their excellent selectivity turns them into attractive proton-conducting membrane materials[6–9] for hydrogen-based technologies. However, those applications require membranes with very high proton (areal) conductivity[10], typically exceeding[11] 5 S cm$^{-2}$, which stimulates research into new 2D proton-conducting materials with higher conductivities than that of graphene and hBN[12,13]. This could be achieved by engineering atomic scale defects[13,14], nanoscale corrugations and strain[5,15] in the existing 2D materials or by growing designer 2D crystals with intrinsic angstrom-scale pores[16–19] (e.g., various graphynes). However, it is challenging to control pores' shape, sizes and other characteristics via these routes, which would allow both high permeability and selectivity. Alternatively, increasing the operation temperature $T$ could in principle lead to an exponential increase in conductivity because proton transport typically involves a finite energy barrier $E$[1,2]. Besides higher conductivity, materials that can operate at elevated temperatures (for example, 200−500 °C) are highly sought after because many chemical engineering and energy conversion applications are more efficient at such $T$. However, the above temperature range—commonly referred to as the proton materials gap[10,12]— remains challenging for both 2D and 3D materials. In addition to their potential in hydrogen-based technologies, proton-permeable 2D materials are also of interest for the use as atomically thin barrier layers for protection and control in catalytic and electrochemical processes[20–22]. In this work, we explore proton transport through titania monolayers[23–25]. The material consists of a 2D array of TiO$_6$ octahedra (Fig. 1a) and inherits its 3D parent's stability in aqueous, oxidizing and reducing environments at elevated $T$. We find an unexpectedly high proton permeability of 2D titania, including at temperatures above 200 °C, and attribute this to an extremely high density of angstrom-scale vacancies.

[1]Institute of Applied Physics and Materials Engineering, University of Macau, Macau, China. [2]State Key Laboratory of Fine Chemicals, School of Chemical Engineering, Dalian University of Technology, Dalian, Liaoning, China. [3]Department of Physics and Astronomy, University of Manchester, Manchester, UK. [4]National Graphene Institute, University of Manchester, Manchester, UK. [5]Institute of Quantum Materials and Physics, Henan Academy of Sciences, Zhengzhou, China. [6]Key Laboratory of Artificial Micro- and Nano-Structures of Ministry of Education, School of Physics and Technology, Wuhan University, Wuhan, China. [7]Research Center for Materials Nanoarchitectonics, National Institute for Materials Science, Ibaraki, Japan. ✉e-mail: guangpinghao@dlut.edu.cn; marcelo.lozadahidalgo@manchester.ac.uk; geim@manchester.ac.uk; pengzhansun@um.edu.mo

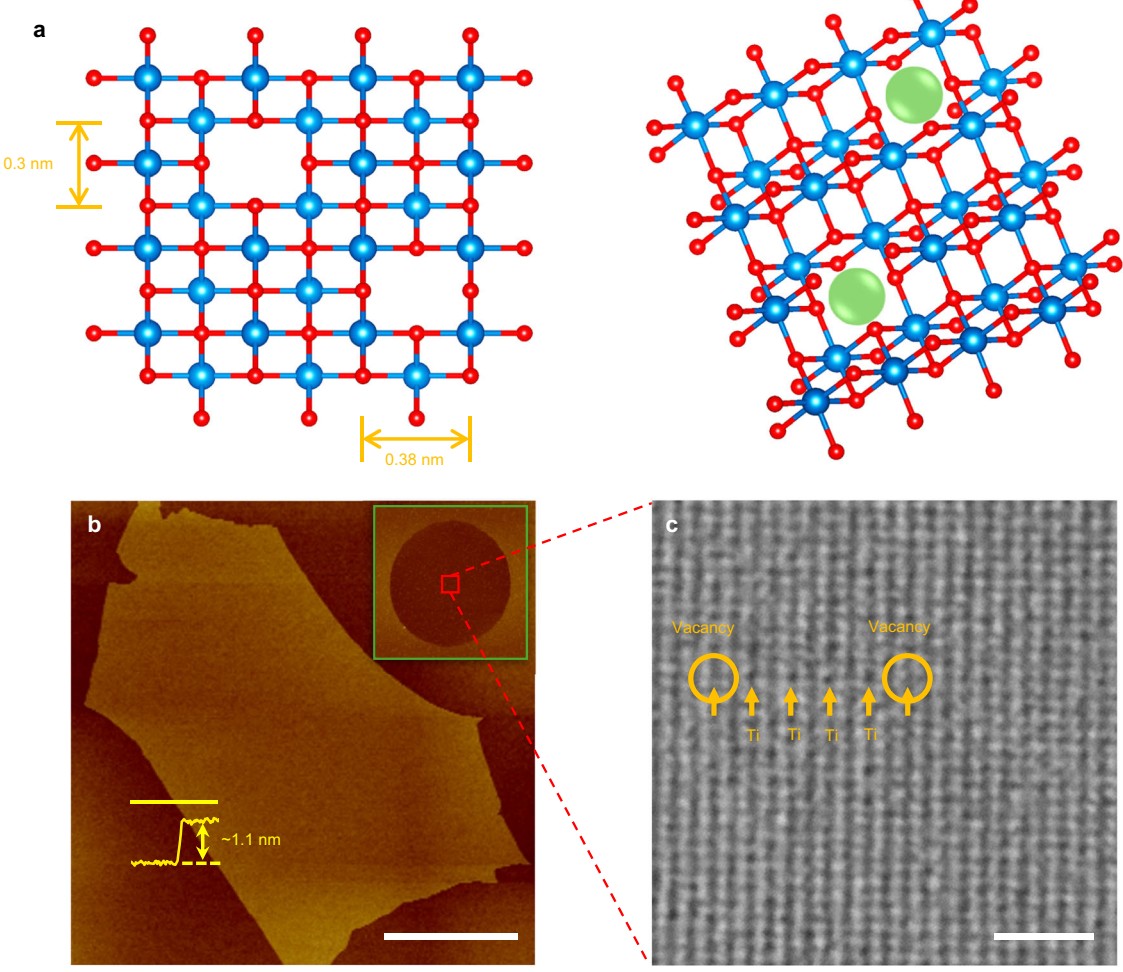

**Fig. 1 | Studied monolayer titania. a** Schematics of single-Ti-atom vacancies in monolayer titania. Left: top view with the shown lattice parameters; right: tilted 3D view. The blue and red balls denote Ti and O atoms, respectively. The green balls highlight vacancy positions. **b** AFM of a titania sheet placed on an oxidized silicon wafer. Scale bar, 10 μm. Yellow profile, height-trace along the yellow solid line indicating the titania thickness. Inset: same crystal after being transferred over a 3 μm diameter aperture etched in a silicon-nitride substrate. **c** HRTEM image of a titania monolayer. Scale bar, 1 nm. Dark spots are Ti atoms. They form an orthorhombic lattice that is often interrupted by brighter blurred rectangles, two of which are indicated by circles.

## Results
### Device fabrication and characterization

The monolayer titania crystals used in this work were prepared by delamination of layered bulk compound $K_{0.8}[Ti_{1.73}Li_{0.27}]O_4$ via ion exchange, following the recipe described previously[23–25]. In brief, the bulk compound consists of titania monolayers with some Ti atoms substituted by Li and the space between the layers is filled with $K^+$ ions, which balance the layers' negative charge. During an ion exchange process, both $Li^+$ and $K^+$ ions were substituted by protons. Then, in the second ion exchange process, the interlayer protons were replaced by large cations $(C_4H_9)_4N^+$ which led to the crystals' delamination and yielded titania monolayers with an effective thickness of ~1.1 nm (Fig. 1b), in agreement with the previous reports[24,25]. Typical lateral dimensions of 2D titania were a few μm but some flakes could reach tens of μm (Fig. 1b and Fig. S1). We searched for the largest monolayers (Fig. S1) and transferred them over apertures of 2-3 μm in diameter etched in silicon-nitride membranes (inset of Fig. 1b and Fig. S2, Supplementary Information), as described previously[1,2]. The resulting freestanding titania monolayers were first examined using atomic force microscopy (AFM), and samples showing cracks, tears or folds were discarded. We then characterized the remaining monolayers under high-resolution transmission electron microscopy (HRTEM). In the HRTEM images (Fig. 1c and Fig. S3), titanium atoms appear as dark spots within an orthorhombic lattice (see the schematic in Fig. 1a). Some of those dark spots were missing, resulting in rectangles with blurred centers (Fig. 1c). According to the previous report[26], this structural feature corresponds to a Ti-atom vacancy. 2D titania has a unit cell of 0.3 nm × 0.38 nm in size (Fig. 1a, left), according to the X-ray diffraction analysis[23], and the space available for proton permeation through a Ti-atom vacancy is only a fraction of the empty space in the ball-and-stick model shown in Fig. 1a because of the dense electronic clouds surrounding the atomic nuclei (Fig. S5 in Supplementary Information). The frequency of these vacancies estimated from the elemental analysis of the material[25] was 13.5%. Our HRTEM images, obtained from a combined area of a few hundred nm² (Fig. S3), yielded a somewhat lower occurrence of ~7.5%, which translates into about one vacancy per nm², or ~10¹⁴ cm⁻². This discrepancy between chemical and TEM analyzes is the same as in the previous work[26]. We attribute it to the fact that our monolayers were selected for their large size and high quality, whereas the elemental analysis was done for macroscopic samples that probably contained flakes with a higher concentration of vacancies. The latter flakes are expected to break more easily and did not survive our selection.

To ensure the absence of nanoscale pinholes (occasionally observed in HRTEM) that could have been missed under the AFM characterization, all our suspended titania monolayer devices were He-leak tested. The test provided a sensitivity down to ~$10^8$ atoms s$^{-1}$ (Supplementary Information, Fig. S4), which would be sufficient to discern gas flows through an individual pore of 1 nm in diameter. In these measurements, one side of the membrane was exposed to helium gas using a maximum pressure of 1 bar. The other side faced a vacuum chamber connected to a He-leak detector. Only two devices were found to exhibit gas flow rates of the order of $10^{13}$ atoms s$^{-1}$ at the 1 bar feed pressure (Fig. S4). Their retrospective examination under a scanning electron microscope revealed a single pinhole of ~50 nm in size (Fig. S4), consistent with the observed Knudsen flow. The leaky devices were excluded from further measurements. All the other membrane devices (20 in total) exhibited no helium leakage, showing that they did not contain even a single one-nm pinhole. The helium tests also demonstrated that numerous vacancies observed in 2D titania by HRTEM are practically impermeable to gases.

### Proton transport

Some of the devices impermeable to helium were then tested for proton permeation. To that end, both sides of the suspended 2D titania were coated with a proton-conducting polymer (Nafion) and electrically connected to proton-injecting electrodes (Pt on carbon), as reported previously[1,2] (inset of Fig. 2a). Typical $I-V$ characteristics for 2D titania at small biases $V \lesssim 100$ mV are shown in Fig. 2a. The current $I$ increased linearly with $V$, which allowed us to extract the areal conductivity $\sigma$. Analysis of several titania devices yielded $\sigma = 2.0 \pm 0.8$ S cm$^{-2}$ (Fig. 2b). This proton conductivity is over 100 times higher than that of monolayer graphene, and more than 10 times larger than for monolayer hBN (Fig. 2a, b). Note that despite the large conductivity, the devices' resistance was still ~2 orders of magnitude higher than that of our reference devices with a bare aperture (no crystal). This confirmed that the measured $\sigma$ was intrinsic to titania monolayers and the series resistance from Nafion was negligible. To gain further insight, we measured how the areal conductivity evolved with temperature. Figure 2c shows that $\sigma$ increased with $T$, following roughly the Arrhenius behavior, $\sigma \propto \exp(-E/k_BT)$. The fitting yields the activation energy $E = 0.34 \pm 0.06$ eV. We attribute the observed high conductivity to the high density of Ti vacancies identified under

HRTEM, a conclusion supported by our theoretical simulations (Fig. S5, Supplementary Information).

The temperature dependence in Fig. 2c suggests that much higher $\sigma$ can be achieved for $T$ inside the proton materials gap[10]. However, Nafion can be used as an electrical contact to titania only over a limited $T$ range because of its dehydration at higher temperatures[11]. For this reason and because of a mechanical strain inflicted on suspended 2D crystals within heated Nafion, we had to limit our $T$ to ~60 °C to avoid their damage, as reported previously[1]. This constraint is, however, not fundamental. Our suspended devices without Nafion could sustain $T$ up to ~260 °C (at higher $T$ they cracked, presumably because of different thermal expansion with respect to the silicon-nitride substrate). Furthermore, our X-ray photoelectron spectroscopy and AFM analyzes revealed that 2D titania retained its crystallographic and chemical structure after being exposed for several hours to 300 °C in various gas atmospheres including argon, air, and hydrogen (Fig. S6).

To measure the proton conductivity at temperatures higher than that allowed by Nafion, we coated the suspended 2D titania on both sides with porous Pt films (~10 nm thick) and placed the devices into a chamber containing humid hydrogen atmosphere (upper inset of Fig. 3). In this configuration, Pt absorbs H$_2$ gas and provides protons for transport through titania[12]. Our measurements using the latter setup are shown in Fig. 3. At room $T$, $\sigma$ was ~3 times lower than that for the Nafion-based devices. We attribute this to either some vacancies being blocked by Pt atoms or a lower proton density on the titania surface covered with Pt (Fig. S7, Supplementary Information). From the measurements at higher $T$, we extracted the activation energy $E \approx 0.36$ eV (Fig. S7, Supplementary Information). Within our accuracy, this is the same $E$ as observed using Nafion devices (Fig. 3 and S7), suggesting the same mechanism governing proton transport in both setups. The areal conductivity reached 100 S cm$^{-2}$ at 200 °C (Fig. 3) and 200 S cm$^{-2}$ at 260 °C (Fig. S7). This is an order of magnitude higher than for the industry standard, Nafion 117, that is, 200 μm thick Nafion films measured at 80 °C (the finite thickness is essential to minimize water and hydrogen permeation)[11] and, also, surpasses the US Department of Energy target (50 S cm$^{-2}$) for proton-conducting materials in hydrogen and fuel cell technologies[27].

We verified that the observed high $\sigma$ did not involve electron tunneling through titania monolayers. To that end, we measured the same devices in vacuum (Fig. 3). At room $T$, no current could be

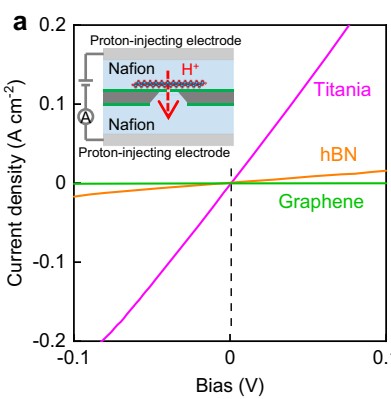
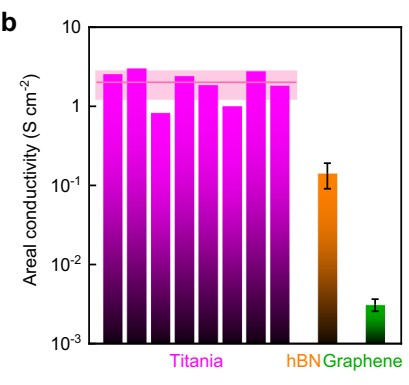
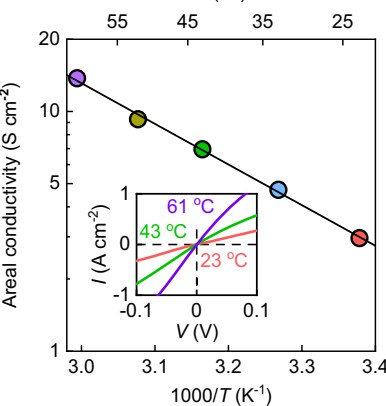

**Fig. 2 | Proton transport through 2D titania. a** Examples of $I-V$ characteristics for titania, hBN, and graphene monolayers (color coded). Inset, schematic of the measurement setup using Nafion as the conducting media. Dashed black line marks zero voltage. **b** Proton areal conductivity of titania devices is compared with that of graphene and hBN monolayers measured using the same setup. For 2D titania, each bar represents a different device. The solid horizontal line marks the average conductivity for all 8 devices with the shaded area indicating SD. For hBN and graphene, the error bars show the average conductivity and SD found from measurements using at least 3 devices. The graphene and hBN data are in quantitative agreement with the previous reports[1,2,5]. **c** Temperature dependence for one of our titania devices. Symbols: experimental data. Solid line: best exponential fit, yielding $E = 0.34 \pm 0.06$ eV. Inset, examples of $I-V$ curves from which $\sigma$ in the main panel was extracted (same color coding as in the main figure).

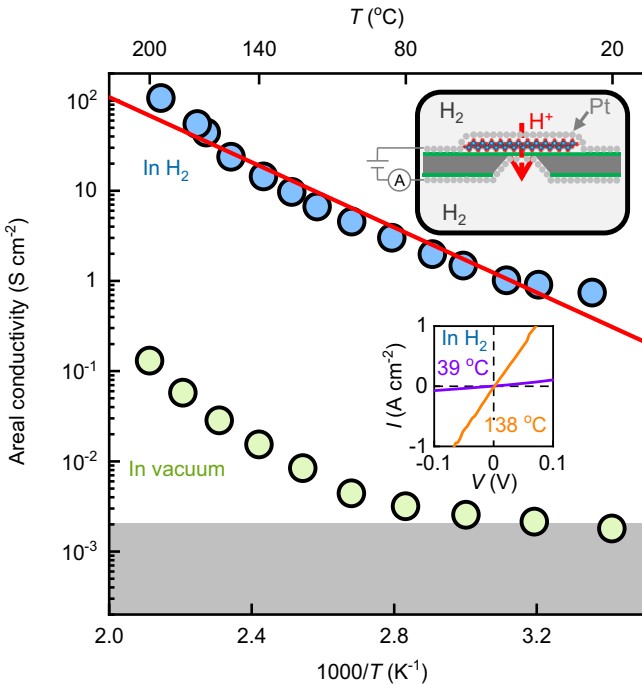

**Fig. 3 | High-temperature proton transport using Pt-coated devices.** Arrhenius plots for the device measured using porous Pt electrodes in H$_2$ gas (1 bar) and vacuum (~10$^{-2}$ mbar), respectively (color coded). Standards deviations (SD) using different devices (not shown) increased with increasing $T$ but did not exceed 50% of the measured values for all temperatures. Solid red line, best exponential fit, yielding $E \approx 0.36$ eV. The gray shaded area marks our lower detection limit. Upper inset, schematic of the experimental setup; lower inset, examples of $I–V$ curves at different $T$ (color coded).

detected within our accuracy of ~10 pA. At higher $T$, the background current started to increase, but $\sigma$ was still three orders of magnitude lower than that in the hydrogen atmosphere for all $T$. This unambiguously corroborates that the high $\sigma$ observed in hydrogen was due to proton conduction and electron tunneling provided a negligible contribution to the overall conductivity, consistent with titania's large bandgap (3.8 eV)[28]. Note also that the slope of the $T$ dependence in vacuum (Fig. 3) was somewhat close to that in the hydrogen atmosphere. This is perhaps unsurprising as not-ultrahigh vacuum systems inevitably contain remnant water adsorbed on surfaces whereas, as shown below, water on 2D titania can serve as a source of protons.

## Ion selectivity

The above experiments show that monolayer titania blocks helium but is highly permeable to thermal protons. In principle, this leaves a chance that small ions like Li$^+$ can also permeate through 2D titania. To assess the latter possibility, we used another experimental setup in which the titania devices separated two reservoirs filled with liquid electrolytes (Fig. 4a, top inset). As a reference, we first filled both reservoirs with HCl solutions and measured the membranes' areal conductivity $\sigma$ using Ag/AgCl electrodes[3]. The conductivity extracted from the linear $I–V$ response was ~1.8 S cm$^{-2}$ for 0.1 M HCl (Fig. 4a). This agrees with $\sigma$ measured for our Nafion-coated devices in Fig. 2, in which Nafion provided a similar proton concentration[1]. In contrast, if we used 0.1 M solutions of KCl or LiCl, $\sigma$ was ~180 times smaller. This clearly indicates that monolayer titania exhibits high selectivity between protons (H$^+$) and other small cations.

We corroborated the high proton selectivity using drift-diffusion measurements (inset of Fig. 4b) which provided information about relative contributions of different ions into the total conductance[3]. To this end, one of the reservoirs was filled with HCl at a relatively high-

concentration ($C_h = 1$ M) and the other one at $C_l = 0.1$ M, which provided the concentration gradient $\Delta C = C_h/C_l = 10$. The measured $I–V$ characteristics using this setup included the well-known contribution due to redox reactions at electrodes[3], which was subtracted from the measure voltage, allowing us to extract the membrane potential $V_m$ (Fig. S8). Figure 4b shows an example of typical $I–V_m$ characteristics found in our drift-diffusion experiments. At zero $V_m$, the current was positive. The concentration gradient drives both H$^+$ and Cl$^-$ from high to low concentration reservoirs, and the positive current unambiguously shows that protons contributed most. The potential drop $V_m^*$, that is required to stop the diffusion current across the membrane, is given by[3] $V_m^* = -(t_H - t_{Cl}) k_B T/e \ln(\Delta C)$ where $t_H$, $t_{Cl}$ are the transport numbers for protons and Cl$^-$ (both numbers are positive and $t_H + t_{Cl} \equiv 1$). Our measurements yielded $V_m^* \approx -59$ mV (Fig. 4b). This translates into $t_H \approx 1$, that is, practically all the current was carried by protons and the Cl$^-$ contribution was small. Our accuracy in measuring $V_m^*$ was ~1 mV as found from several replicated measurements. We performed similar drift-diffusion experiments using LiCl and KCl solutions and, as expected, the observed $V_m^*$ was close to zero within the same accuracy.

This left the question of where the small but clearly discernable areal conductivity $\sigma_0$ observed for KCl and LiCl solutions came from (Fig. 4a). Indeed, $\sigma_0$ was close to 10$^{-2}$ S cm$^{-2}$, nearly an order of magnitude larger than our detection limit, whereas the finite accuracy of the drift-diffusion experiments might still allow minute flows of ions through 2D titania membranes. To address the above question, we performed additional experiments and found that $\sigma_0$ did not depend on KCl and LiCl concentrations (Fig. 4c) and only slightly changed if we utilized other chlorine solutions (MgCl$_2$, CaCl$_2$ and Ru(bipy)$_3$Cl$_2$; Fig. 4d). Moreover, the same areal conductivity $\sigma_0$ was found for deionized water (Fig. 4c). This shows that the observed $\sigma_0$ cannot be attributed to ions. Their translocation through single-atom vacancies should be blocked because of the relatively large diameter of hydrated ions. However, we also cannot rule out a role of hydrocarbon contamination that is practically unavoidable for surfaces prepared in air, not under ultra-high vacuum conditions. Hydrocarbon molecules would then be expected to reduce the space available for ion passage but to be less detrimental for proton transport. In either case, the observed $\sigma_0$ can be attributed to residual protons that are always present in water. Indeed, although the bulk conductivity of deionized water is insufficient to account for the observed value of $\sigma_0$, note that the titania surface is well known for its water dissociative properties[29] and our titania monolayers carried a large negative charge (Fig. S9, Supplementary Information). Accordingly, this should have resulted in a high density of protons adsorbed on titania membranes, which could then diffuse along the surface and transfer through vacancies, giving rise to the finite $\sigma_0$ observed for all aqueous solutions. Further work is required to understand the reason for the finite conductivity observed for salt solutions.

To provide more information about the proton transport through monolayer titania, we studied the isotope effect. To this end, deuterium chloride (DCl) dissolved in heavy water (D$_2$O) was used and compared with HCl in H$_2$O for the same range of concentrations (Fig. 4c). We observed qualitatively the same dependence of 2D titania's areal conductivity as a function of H$^+$ and D$^+$ concentrations. However, $\sigma$ for deuterons D$^+$ was lower than that for protons H$^+$ by a factor of $1.6 \pm 0.16$ (using the linear fits for the high-concentration regime in Fig. 4c). The isotope effect clearly corroborates that the observed conductance was indeed due to protons. Our theoretical analysis for the observed $E$ and the D$^+$/H$^+$ separation factor suggests that protons first attach to broken bonds of Ti-atom vacancies and then translocate through the 2D crystal. The broken bonds at the vacancy edges make the titania membrane highly negatively charged, as evidenced by the zeta potential measurements (Fig. S9), which consequently attracts a high density of protons. When a voltage bias is

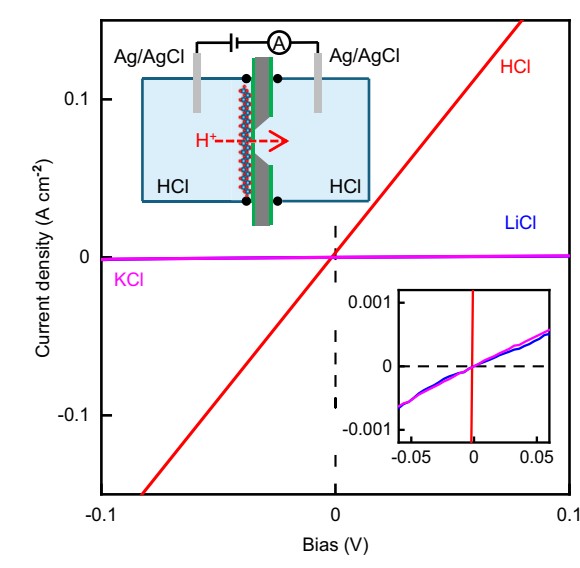

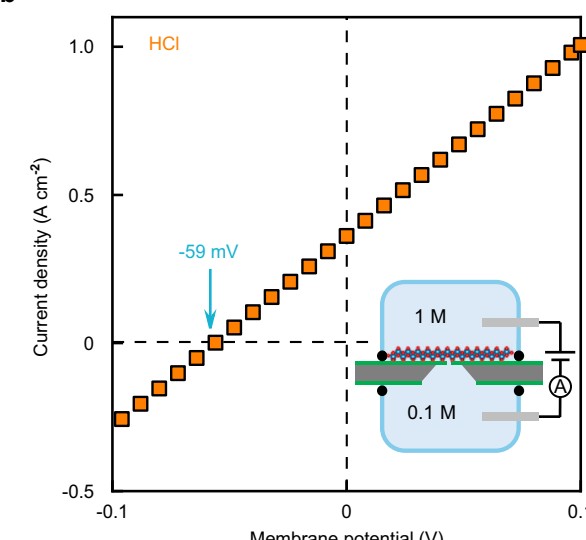

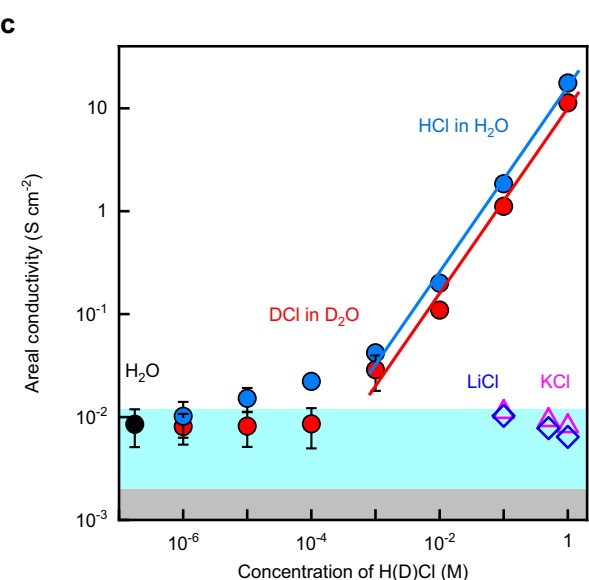

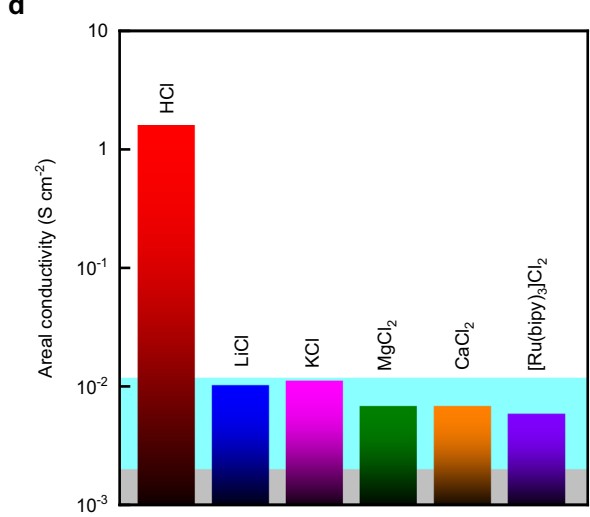

**Fig. 4 | Ion selectivity measurements. a** Examples of $I$–$V$ characteristics for HCl, KCl and LiCl (color coded). Bottom inset, zoom-in. Top inset, schematic of the experimental setup. **b** Example of $I$–$V$ characteristics in the drift-diffusion experiments using HCl in concentrations 0.1 and 1 M as illustrated in the bottom inset. The blue arrow indicates the membrane potential of −59 mV that corresponds to the perfect proton selectivity with respect to Cl ions. **c** Concentration dependences using HCl dissolved in $H_2O$ and DCl in $D_2O$ (color coded). Symbols, data taken in 5 different measurements with the error bars indicating SD (shown if larger than the symbols). Solid lines, best linear fits at high concentrations. Also shown are $\sigma$ for LiCl and KCl solutions at different concentrations and for deionized water (color coded). **d** Conductivities for various 0.1 M salt solutions (color coded; SD were less than 40% of all the solutions). The gray areas in (**c**, **d**) mark our detection limit because of leakage currents. Blue areas, $\sigma$ measured using deionized water.

applied across the membrane, protons from the surrounding media (electrolytes or Pt films) are injected into the crystal. Our calculations (Fig. S5) show that protons can then hop between oxygen atoms along the vacancy, leading to a proton current. This resembles proton hopping along water chains in the Grotthuss mechanism known for bulk water, except that in our case water molecules are replaced with oxygen bonds in titania pores (Supplementary Information provides comparison of the inferred process with those in known biological and solid-state 1D channels).

## Outlook

Our experiments show that protons can permeate through monolayer titania crystals whereas helium atoms are excluded. At room

temperature, the observed areal conductivity of protons in monolayer titania is orders of magnitude higher than that of graphene and hBN monolayers. The titania conductivity exceeds 100 S cm$^{-2}$ at 200 °C, making it an attractive proton-conductive material within the infamous proton materials gap[10]. In principle, titania monolayers can be prepared via scalable routes involving soft-chemistry procedures[23–25] and assembled over large areas to form quality membranes via techniques such as layer-by-layer electrostatic assembly and Langmuir–Blodgett deposition[30,31]. Furthermore, the density of monovacancies in 2D titania can be changed if required from ~9% up to ~18% using different compositions of the original bulk compound used for exfoliation[32,33]. Not only titania but also other 2D oxides can potentially be used as membranes, separators and protective coatings

in renewable energy applications such as fuel cells, electrolyzers and catalytic systems where rapid proton transport combined with gas and ion impermeability is essential.

## Methods

### Device fabrication

We followed the well-established soft-chemistry procedures[23–25] to prepare monolayer titania crystals. In brief, layered titanate compounds $K_{0.8}[Ti_{1.73}Li_{0.27}]O_4$ were obtained by mixing potassium carbonate ($K_2CO_3$, Sigma–Aldrich), lithium carbonate ($Li_2CO_3$, Sigma–Aldrich) and titanium dioxide ($TiO_2$, rutile form) according to a molar ratio 2.4:0.8:10.4, followed by decarbonating at 800 °C for 0.5 h and a further calcination at 1100 °C for 20 h. The products were stirred vigorously in 1 M of HCl solution for a few days so that the interlayer potassium ions and intralayer lithium ions were fully extracted and exchanged for protons, resulting in protonic compounds $H_{1.07}Ti_{1.73}O_4$ as determined by chemical analysis. To delaminate for monolayers, the material was dispersed in a tetrabutylammonium hydroxide [$(C_4H_9)_4NOH$] aqueous solution and mildly shaken for 10 days, resulting in 2D titania $Ti_{0.87}O_2^{0.52-}$.

The 2D crystals were casted onto a freshly cleaned oxidized silicon wafer and then checked under an optical microscope. Figure S1a shows that the typical lateral dimension of the crystals was a few μm and some reached a few tens of μm. These large flakes were carefully examined under dark field and differential interference contrast (DIC) modes (Fig. S1b, c) to ensure they were in high quality. Only those free from any contaminations, wrinkles, cracks and other imperfections were chosen for device fabrication. They were transferred over an aperture 2-3 μm in diameter that was microfabricated in a silicon-nitride (SiNx) chip (500 nm thick) using the technique standard for van der Waals assembly[34,35]. Details for making the SiNx microapertures were well-documented previously[1,2] and are schematically illustrated in Fig. S2.

To make the Nafion-coated device for proton transport measurements, we carefully drop-casted Nafion (Sigma–Aldrich, 5 wt% 1100EW) solution on both sides of the fabricated device (step 5 of Fig. S2), followed by electrically connecting to a pair of proton-injecting electrodes (Pt on carbon) (inset of Fig. 2a). The assembly was baked at 130 °C under 100% relative humidity to crosslink the Nafion monomers so that the resulting polymer layers were highly proton-conductive but remained electron-insulating. For measurements at elevated temperatures $T$, instead of casting Nafion, we sputtered porous Pt films (a few to tens of nm thick) on both sides of the device following the procedures developed previously[12] (step 6, Fig. S2). These Pt films served as both electrodes and proton reservoirs in a humid hydrogen atmosphere.

### HRTEM imaging

The exfoliated titania monolayers were characterized using a transmission electron microscope (JEOL JEM-ARM200F) which was equipped with an image corrector of corrected electron optical systems (CEOS). To remove hydrocarbon and other contaminations on the crystals' surface, they were first UV-treated for 2 h and then baked for another 2 h at 150 °C. The acceleration voltage was set to 80 kV and the current density was ~1.7 pA/cm$^2$. To image the atomic structure, a maximum magnification of 800,000 times was used. Figure 1c shows one of our obtained HRTEM images and a larger view is provided in Fig. S3. They were captured using a Gatan OneView camera under an exposure time of 6.5 s with drift-correction.

### Helium leak tests

In addition to the extensive AFM imaging of the transferred titania monolayers, we also performed helium leak tests for all our devices to check for any nm-scale pinholes and other imperfections. In those measurements, the tested device separated two vacuum chambers (inset of Fig. S4). One of them (feed chamber) was connected to a

helium-gas reservoir. The injection of helium gas was electrically controlled by a dosing valve and its pressure was recorded by a pressure gauge. The other (permeate chamber) was connected to a leak detector (*Leybold L300i*). Its sensitivity limit with respect to helium flows was of the order of ~10$^8$ atoms s$^{-1}$, as determined by control measurements using a piece of bare silicon wafer. This sensitivity allows discerning Knudsen flows through a single pinhole down to 1 nm in size. Prior to real tests, the setup was sealed and each chamber was leak-checked to ensure that the only possible gas pathway between the two chambers was through the device.

### Electrical measurements for the transport of protons and ions

To measure the transport of protons through 2D titania, Nafion-coated devices were placed inside a chamber filled with 1 bar of $H_2$ at 100% relative humidity. The $I$–$V$ characteristics were measured using the source meter *Keithley* 2636B and collected using software *LabVIEW*. Normally, we limited the applied voltage to ≲100 mV to ensure linear response and at a sweep rate of 5 mV s$^{-1}$. The measurement temperature $T$ was limited to 60 °C to avoid dehydration in the Nafion films. We note that according to the previous reports[32,33], the porosity of 2D titania can be tuned from ~9% to ~18% if required by using layered precursors of different compositions and in principle, we would expect a porosity dependence of the areal conductivity using samples having different porosities. However, our experimental sensitivity in proton transport experiments ($2.0 \pm 0.8$ S cm$^{-2}$, Fig. 2b) was about 40%, which is greater than the described porosity range. Such variations are therefore not expected to result in a measurable difference in the areal conductivity of protons. For higher $T$ measurements such as those in Fig. 3 and S7, Pt-coated devices were used instead and all the other conditions remained the same.

To find out whether or not ions could permeate through the titania monolayers, we employed a customized setup (inset of Fig. 4a) which consisted of two reservoirs separated by the fabricated device. Prior to measurements using different salt solutions, the reservoirs were first flushed with an isopropanol/water mixture (1:1 in volume) and then deionized water to ensure proper wetting of the membrane's surface. The following solutions were tested: HCl, LiCl, KCl, $MgCl_2$, $CaCl_2$, and $[Ru(bipy)_3]Cl_2$. Each of them was carefully introduced into the two reservoirs simultaneously and Ag/AgCl electrodes were inserted for electrical measurements. All these experiments were done at room $T$ ($297 \pm 3$ K).

### Density functional theory calculations

To provide theoretical insights for the observed proton transport, we performed density functional theory (DFT) calculations using the VASP package[36]. The exchange-correlation potential and ion-electron interactions were described using the generalized gradient approximation (GGA) and projected augmented wave (PAW) methods[37,38]. A kinetic energy cutoff of 500 eV was employed. The van der Waals interactions were addressed by the semi-empirical DFT-D2 method[39,40]. All atoms were allowed to fully relax to the ground state by taking into account the spin-polarization. The relaxation resulted in the optimized lattice constants of 3.77 Å and 3.03 Å, respectively, for the 2D titania crystal. Then a single-Ti-atom vacancy was created in the 2 × 3 supercell. This resulted in eight under-coordinated oxygen atoms bonded to the edge of the vacancy: two of them coordinated to a single-Ti atom; another two coordinated to two Ti atoms and the rest to three Ti atoms. As per the previous study[26], the vacancy model of removing one Ti atom together with two single-bonded O atoms well reproduced the HRTEM imaging results because those O atoms are most reactive and tend to desorb from the vacancy. This model was adopted in our simulations. Due to the other unsaturated O atoms, the vacancy was negatively charged. This is in qualitative agreement with our zeta potential measurements (Fig. S9, details see below), showing a large negative surface charge under low proton concentrations. Comparing with the

lattice constants for an intact titania crystal, the vacancy model exhibited slightly larger interatomic separations of 3.97 Å × 3.38 Å. Nonetheless, the space available for proton permeation should be smaller, due to the electronic cloud surrounding the atomic nuclei, as shown by the electron density calculations (Fig. S5a) and in the HRTEM image (Fig. 1c). To simulate the transport process of protons through the vacancy, we first put a proton at infinity and allowed it to move toward the vacancy under the electrostatic attraction from the negatively charged surface. The pathway was fixed perpendicularly to the crystal's basal plane. After crossing through the vacancy, the proton was forced to move away from the crystal by overcoming the latter's electrostatic resistance. For comparison, we also simulated the same proton transport process but through an intact titania lattice. The transition states were searched using the climbing-image nudged elastic band (CINEB) method[41,42].

## Thermal stability

To assess the thermal stability of 2D titania, we employed X-ray photoelectron spectroscopy (XPS) and AFM to characterize its chemical and crystallographic structures after thermal cycling to a higher $T$ (up to 300 °C). To this end, the material was deposited on an oxidized silicon wafer and then annealed at 300 °C for 3 h. Different gas atmospheres were used for the annealing: air, 1 bar of $H_2$ and 1 bar of Ar, respectively. The treated material was carefully characterized using XPS (ESCALAB 250Xi, Thermo Fisher Scientific) equipped with Al Kα X-rays ($h\nu = 1486.7$ eV). To quantify the composition and relative amount of Ti- and O-containing groups/bonds, the XPS Ti 2p and O 1s spectra were analyzed and fitted by Gaussian–Lorentzian functions.

## Zeta potential

To provide information about the surface charging state of 2D titania, we measured zeta potential ($\zeta$) for an aqueous colloidal suspension containing delaminated titania monolayers. Zeta potential is established in all solid-electrolyte systems and characterizes the potential at the slipping surface outside the stationary Helmholtz layer. It is determined by the surface charge density $\rho_s$ and the concentration $C$ of electrolyte. It is well known that $\rho_s$ is also sensitive to the concentration of protons (that is, solution pH) because the present protons easily adsorb on the surface and tune its $\rho_s$. For this reason, we should expect a strong dependence of $\zeta$ on the solution pH. To seek for this effect, we added HCl into the titania suspension to adjust its pH. This setup ensures the same chemical environment as in the ion selectivity measurements (Fig. 4) where HCl solutions of different $C$ were also measured.

## Data availability

All data supporting the key findings of this study are available within the article and the Supplementary Information file. All raw data generated during the current study are provided in the Source Data file. Source data are provided with this paper.

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

## Acknowledgements

This work was supported by the Science and Technology Development Fund (FDCT), Macao SAR (0063/2023/RIA1), the Natural Science Foundation of China (NSFC, 52322319), UM research grant (SRG2022-00053-IAPME), UM and UMDF research grant (MYRG-GRG2023-00014-IAPME-UMDF), the European Research Council (grant VANDER), the Lloyd's Register Foundation (grant Designer Nanomaterials), UKRI (EP/X017745: M.L.-H), the Royal Society (URF\R1\201515: M.L.-H.) and Directed Research Projects Program of the Research and Innovation Center for Graphene and 2D Materials at Khalifa University (RIC2D-D001: M.L.-H. and A.K.G.).

## Author contributions

P.Z.S., A.K.G., M.L.-H. and G.-P. H. designed, directed the project and analyzed the results. Y.J., P.Z.S., Y.L. and W.Z.Z. fabricated the titania devices. Y.J. and G.-P. H. performed proton and ion transport measurements. Y.-T.T. performed gas permeation measurements. W.Q.X. and S.J.Y. performed DFT calculations. R.Z.M. and T.S. synthesized titania sheets. D.-M. T. performed HRTEM characterizations for the titania samples. P.Z.S., A.K.G. and M.L.-H. wrote the manuscript with input from all authors.

## Competing interests

The authors declare no competing interests.
