## [Transparent Peer Review file · Nature Communications]

High proton conductivity through angstrom-porous titania

Corresponding Author: Professor Pengzhan Sun

Version 0:

Reviewer comments:

Reviewer #1

(Remarks to the Author)

The paper by Ji et al. reported high proton conductivity through an angstrom-porous titania 2D crystal whereas small ions and even helium gas atoms were completely blocked within the measurement accuracy. The chosen material having a high density of single-atom vacancies of uniform sizes is interesting and in the referee's opinion, is more feasible in terms of practical membrane applications than those prepared by the widely discussed defect creation methods. The results and analysis are interesting and self-consistent. Specially, the measurements on high-temperature proton conductivity (more than 100 S per cm squared at temperatures over 100 degree C) are impressive, which fell into the infamous proton materials gap and may trigger future research and technology development from a wider range of community. Therefore, the referee recommends for publication in Nature Communications after addressing the following comments.

For the statistical analysis of vacancy density from HRTEM image in Fig. S3, the authors randomly chose a few areas and counted separately the number of single-atom vacancies with respect to the total number of unit cells, giving each area an occurrence frequency. The average value for all areas was used as an estimation for the final frequency. Because multiple areas were analyzed, the referee suggests use of mean plus standard deviation to make the statistics more accurate and rigorous.

In Fig. S5, the energy profile calculated for the transport of protons through a single-atom vacancy in 2D titania is a bit asymmetrical. Although the referee understands that the authors emphasized a qualitative agreement between the experiments and simulations, it is suggested that the authors explain the origin of this asymmetry. To move a step further, the calculated energy barriers (the step between the peak and the minima) from both sides are slightly higher than that obtained from experiments, the authors are suggested to provide at least a qualitative explanation for this difference.

The single-titanium vacancies were introduced spontaneously during the preparation of 2D titania sheets in water. Their density was reported to be determined by the amount of titanium atoms being substituted by lithium atoms in their bulk layered precursors. In this context, is it possible to tune the density of single vacancies by, for example, tuning the composition of the layered precursors? If so, this would definitely add more promise for applications as sieving membranes, as compared to those porous membranes prepared by top-down methods, say, etching or bombardment, because the latter membranes are suffering from a wide pore size distribution.

The authors reported careful proton transport measurements through an individual 2D titania crystal. The referee understands that from a fundamental viewpoint, the results demonstrating a high proton conductivity through the material and complete blockage of other ion and gas species are new and novel to the community. The presented experimental results are already sufficient for publication. Nonetheless, in terms of practical applications for, say, proton conductors, large-area membranes are needed. The authors are suggested to discuss potential ways of extending the results obtained from a single crystal to a larger scale, for example, any possible strategies to prepare large-area membranes for future research.

Reviewer #2

(Remarks to the Author)

In this work, the authors show that monolayer titania is highly permeable to protons while remaining impermeable to helium and small ions including Cl^- and Li^+ . There are still some questions needed to be elucidated before further consideration.

1. How to control the number and distribution of titanium atom vacancies on each monolayer of titania. This irregularity may lead to differences in conductivity between different samples.
2. In HRTEM images, the size of titanium atom vacancies is about 0.1-0.2 nm, which does not correspond to the simulated vacancy size of 0.3-0.4 nm. It is recommended to supplement with single-crystal analysis.

3. During the preparation process, an ion exchange process was used to replace lithium and potassium ions with protons to achieve delamination. This indicates that mobile cations and protons remain in the system, raising the question of how to exclude the influence of these cations. Due to the extremely low thickness of the titania monolayer, even trace amounts of cations and protons could cause significant deviations in the conductivity measurements.
4. Although the short-term thermal stability of titania has been confirmed, there is a lack of research on its performance under long-term high-temperature exposure. This is crucial for evaluating the material's lifespan in practical applications. For example, titania has two main crystalline forms: anatase and rutile. Especially at high temperatures, anatase tends to convert to rutile.
5. Titania is a good electronic conductor. It is recommended to include ion migration number tests to distinguish between the contributions of ion and electron conductivity.
6. The proton conductivity of pure water at room temperature is very low, about $5.5 \times 10^{-5} \mu\text{S}/\text{cm}$. The measured conductivity of KCl and LiCl solutions is about $10^{-2} \text{ S}/\text{cm}$, indicating that lithium and potassium ions can permeate through the titania monolayer.
7. Given that many two-dimensional nanosheets have been proven to conduct protons, it is expected that titania monolayers containing angstrom-scale titanium atom vacancies can be used to conduct protons. It is necessary to clarify the novelty of the paper and provide practical application tests to demonstrate its potential.
8. Two-dimensional crystal materials such as graphene and hexagonal boron nitride (hBN) can effectively sieve hydrogen isotopes. These one-atom-thick crystals are impermeable to atoms and molecules but allow hydrogen ions (thermal protons) to penetrate. How to prove that the proton conduction mechanism of titania monolayers differs from that of graphene and hBN? Does proton transport occur through titanium atom vacancies?
9. Generally, the proton conductivity of materials is affected by spatial hindrance. The pore size of the titania monolayer is less than 1 \AA , so obtaining proton conductivity likely requires very high temperatures, making it difficult to detect proton conductivity at low temperatures.

Reviewer #3

(Remarks to the Author)

This work reports high proton conductivity through angstrom-porous titania membranes. The authors prepared 2D titania, i.e., 2D array of TiO₆ octahedra, and utilized the high density of angstrom-scale Ti-atom vacancies for proton transport across the 2-D membrane. After carefully characterizing the titania membrane, the authors first measured proton transport at low temperatures (i.e., from room temperatures to 60 °C) using a Nafion-based device where the 2-D membrane is coated with Nafion and proton-injecting electrodes (Pt on carbon) are used for electrical connections. They measured a proton conductivity of 2 S/cm² for the titania membrane at room temperature which is one or two orders of magnitude higher than that for other 2-D membranes and found the activation energy for proton transport is 0.34 eV. The authors then measured the membrane proton conductivity at temperatures ranging from 20 to 260 °C using a Pt-coated device where porous Pt films are coated on both sides of the titania membranes as the proton source and the electrodes. They found that the membrane proton conductivity reaches 200 S/cm² at 260 °C and the activation energy measured from the Pt-coated device is very similar to that measured from the Nafion-based device. Finally the authors used an aqueous system where two sides of the membrane were filled with aqueous solutions and Ag/AgCl electrodes were used for electrical connections to demonstrate the excellent proton selectivity over other ions.

Overall, this work is a complete and excellent work. The high proton conductivity of the 2D titania membrane could really make it a good proton conductive material within the well-known proton materials gap, holding great promise for a variety of hydrogen-based technologies. The paper is well-written and the proton transport measurements using the three different systems were nicely designed and performed. Personally, I think this work meets the requirements of Nature Communications. However, there are still some problems related to the claims and the explanations. I would like to ask the authors to address and/or clarify those issues before recommending its publication in Nature Communications. Please find my detailed comments below.

1. The proton conductivity of the titania membrane measured from the Nafion device is quite different from that measured from the Pt coated device. The authors attributed that to possible Pt blockage of the vacancies or the lower proton density on the surface of titania for the Pt coated device. For the second possible explanation, is it possible to estimate the proton density inside the porous Pt thin film at the experimental conditions? It would be great if the authors can provide such an estimation and compare it with the proton concentration (~0.1 M) inside the Nafion film.

2. The analysis and discussion for the drift-diffusion experimental result is not correct. The total diffusion voltage measured at zero current should include contributions both from the selective transport and the electrode redox reaction. $V = V_{\text{transport}} + V_{\text{electrode}} = -(t_{\text{H}} - t_{\text{Cl}}) * k_{\text{B}} * T / e * \ln(\Delta C) - k_{\text{B}} * T / e * \ln(\Delta C)$ according to the authors' measurement configuration. However, the authors totally ignored the second term in their analysis and discussion. As the second term is -59 eV for a Cl⁻ concentration gradient of 10 across the membrane, the first term, $V_{\text{transport}}$ is actually close to zero, indicating that the titania membrane has no ion selectivity at all for concentrations above 0.1 M. I do not understand this result since the conductivities of 1 M KCl and LiCl are four orders of magnitude lower than the conductivity of 1 M HCl and thus the membrane seems to only allow proton transport even for 1 M solutions. The authors should explain this conflict between the drift-diffusion result and the ion selectivity result. Also, I would suggest that the authors re-conduct the drift-diffusion experiments but change the HCl concentrations on two sides of the membranes to 0.1 M and 10 mM or to 10 mM and 1 mM respectively. They may be able to use these new results to better show the ion selectivity if surface-charge-governed ion transport plays an important role on the ion selectivity.

3. The authors claimed that the activation energy measured from the Nafion device and the Pt devices is almost the same and thus the transport mechanism is the same for both type of devices. However, the Arrhenius plots for the Pt-coated devices (Fig. 3 and Fig. S7) are not that linear. In particular, the conductivities measured at higher temperatures deviate from the linear fitting, indicating possibly a larger activation energy for higher temperatures. The authors should try to explain why there is a nonlinear temperature dependence and why the activation energy would increase at higher temperatures.

4. While the authors performed a thorough experimental study to show fast and selective proton transport across the titania membrane, they only provide a very short discussion/explanation about the corresponding mechanism at the end of the manuscript. I would suggest that the authors expand the mechanism discussion in the revised manuscript. Specifically, How large is the measured activation energy when compared with those measured in other small channels or pores? (e.g., J Shao et al. Nature Communications 2015 and R. Tunuguntla et al., Nature Nanotechnology, 2016)? How do protons transport across the membrane? Is it based on the Grotthuss mechanism or regular vehicle mechanism? The authors should consider moving part of the DFT result discussion in the supplementary information to the main text to better explain the mechanism. Also, it would be interesting to compare the proton conductance of a single titanium vacancy with those from natural or artificial small channels/pores, e.g., gramicidin A channels and sub-1 nm carbon nanotube porins. Would the 2D titania vacancy exhibit much higher proton conductance?

5. When discussing the proton conductivity at elevated temperatures, the authors compared the conductance of an angstrom thick titania membrane with a 200- μm -thick Nafion membrane. This comparison seems unfair because of the significant difference in membrane thickness. Thinner Nafion membranes would definitely exhibit much higher conductance. I believe that the reason why industry choose the 200- μm -thick Nafion membrane is more from a durability perspective (including both mechanical and chemical stability). Will the 2-D titania membrane provide similar durability as the thick Nafion membrane? I would suggest that the authors add a short discussion on this to better justify the titania membrane will have promising applications for hydrogen-based technologies. Also, the authors should do a better job in justifying why we need 2D membranes as proton conducting membrane materials for hydrogen-based technologies at the beginning of the manuscript. Is it mainly because of the proton materials gap? Nevertheless, the mechanical stability of 2-D membranes will always be an issue for practical applications.

6. Fig.4c clearly shows that the proton concentration inside the 2-D membrane depends on the surrounding medium concentration and there is a surface-charge-governed proton transport regime. It is great to see that the authors did zeta potential measurement for the titania surface and estimate the surface charge density. I would also suggest that the authors estimate the corresponding proton concentration inside the membrane below which surface charges dominate the proton transport using the expression $c = 4 \cdot \sigma / \text{diameter} / e / 1000 / N_A$. They should check if this estimation can explain the observed HCl concentration dependence observed in Fig. 4c.

7. There are no error bars for the conductivity data in Fig. 2c. Also, personally, I do not like the grey "representative" symbols with the error bars in Fig.3 and Fig. 4d. Since the measured conductivity varies from 0.01 to 100 S/cm², how could a single symbol in each plot represent all error bars? I would suggest that the authors add error bar for each data point in those two figures.

Reviewer #4

(Remarks to the Author)

The authors present a study of ion and gas transport through freestanding titania monolayers supporting atop micrometer-sized holes in silicon nitride supports, with each side of the layers accessible to varying solutions and/or gases. They find that titania layers free of any large mechanical defects are excellent barriers to all species except proton over a very wide temperature range, from ambient conditions where protons exist principally as hydronium ions, to >200 C where protons lack solvation. Areal conductances are quite high for protons and are orders of magnitude lower for other ions.

The work extends prior work from the same group on proton conduction through other 2D materials, notably graphene and boron nitride. The work is well executed with many control experiments to rule out potential artefacts from macroscopic defects. The lack of any gas transport in He leak tests, the clear and repeatable difference between apertures with and without titania layers, the near-zero conductance in the absence of hydrogen, and the expected trends in cell potential for solutions of differing HCl concentration on the two sides of the membrane, all are consistent with their interpretation of high proton conductance and high proton transference through titania layers. Their statements regarding the possible role of proton attachment to broken bonds of Ti-atom vacancies is sensible and consistent with the similarity of mechanisms over the wide temperature range which includes conditions where protons are primarily solvated, and where they are not solvated.

I have only two minor comments. First, the manuscript is inconsistent in usage of the term "conductivity", with the term in some places being correctly identified as areal conductivity (i.e., conductance normalized by area), and in some places the simple term "conductivity" is used. Conductivity is a bulk property and has units of S cm⁻¹; that is not how the term is used here. I recommend that in all instances where the term "conductivity" is used, that the term is modified to always state that it really means "areal conductivity". In this way, confusion with other studies that do truly report bulk conductivity will be avoided.

The second comment is more general and has to do with the possible effects of hydrogen gas on titania properties. Titania normally has a Ti valence of 4+ which makes it a wide-bandgap insulator, but contact with hydrogen gas can reduce some

Ti atoms to the 3+ valence state, which then makes the material mixed-valent and a potential electronic conductor. These changes could be highly susceptible to environmental exposure and could result in a situation where the titania layers become doped on exposure to hydrogen, and then de-doped as hydrogen is removed. I'm not sure what the best test for this would be, but I present the possibility as something for the authors to consider.

Version 1:

Reviewer comments:

Reviewer #1

(Remarks to the Author)

The authors have addressed all concerns and the manuscript has been improved significantly. Now I can recommend its publication.

Reviewer #2

(Remarks to the Author)

Most issues have been addressed. It can be accepted.

Reviewer #3

(Remarks to the Author)

The authors have done an excellent job in addressing my comments in the revised manuscript. I therefore recommend publishing this paper in Nature Communications

Reviewer #4

(Remarks to the Author)

The authors have adequately addressed my comments from earlier review, particularly involving issues associated with mixed valence in partially reduced titania.

RESPONSE TO REVIEWERS' COMMENTS

Reply to comments of Reviewer #1

The paper by Ji et al. reported high proton conductivity through an angstromporous titania 2D crystal whereas small ions and even helium gas atoms were completely blocked within the measurement accuracy. The chosen material having a high density of single-atom vacancies of uniform sizes is interesting and in the referee's opinion, is more feasible in terms of practical membrane applications than those prepared by the widely discussed defect creation methods. The results and analysis are interesting and self-consistent. Specially, the measurements on high-temperature proton conductivity (more than 100 S per cm squared at temperatures over 100 degree C) are impressive, which fell into the infamous proton materials gap and may trigger future research and technology development from a wider range of community. Therefore, the referee recommends for publication in Nature Communications after addressing the following comments.

We thank the Reviewer for this positive assessment of our work.

For the statistical analysis of vacancy density from HRTEM image in Fig. S3, the authors randomly chose a few areas and counted separately the number of single-atom vacancies with respect to the total number of unit cells, giving each area an occurrence frequency. The average value for all areas was used as an estimation for the final frequency. Because multiple areas were analyzed, the referee suggests use of mean plus standard deviation to make the statistics more accurate and rigorous.

We have implemented this suggestion (page 2 of SI). The vacancy density is estimated as $(7.5 \pm 0.4)\%$ on the basis of six areas shown in Fig. S3, each about 25 nm^2 in size.

In Fig. S5, the energy profile calculated for the transport of protons through a single-atom vacancy in 2D titania is a bit asymmetrical. Although the referee understands that the authors emphasized a qualitative agreement between the experiments and simulations, it is suggested that the authors explain the origin of this asymmetry. To move a step further, the calculated energy barriers (the step between the peak and the minima) from both sides are slightly higher than that obtained from experiments, the authors are suggested to provide at least a qualitative explanation for this difference.

The observed asymmetry in the energy profile (Fig. S5c) can be attributed to the asymmetric structure of the single-Ti-atom vacancy. As shown previously [M. Ohwada et al, *Sci. Rep.* **3**, 2801 (2013)], removing a single Ti atom from a titania lattice results in eight under-coordinated oxygen atoms bonded to the edge of the vacancy: two of them coordinated to a single Ti atom; another two coordinated to two Ti atoms and the rest to three Ti atoms. The two single-bonded oxygen atoms are most reactive and tend to desorb from the vacancy, leaving the rest oxygen atoms forming an asymmetric structure, as schematically shown in the insets of Fig. S5c.

On the other hand, the difference between the calculated (0.4 – 0.6 eV) and measured energy barrier (0.3 – 0.4 eV) can be tentatively attributed to the presence of nanoscale corrugations in monolayer titania membranes. Suspended 2D crystals are never perfectly flat because of thermal fluctuations (flexural phonons) and local strain, which generate nanoscale ripples [J. C. Meyer et al, *Solid State Commun.* **143**, 101–109 (2007); A. Fasolino, et al, *Nat. Mater.* **6**, 858–861 (2007); R. Zan et al, *Nanoscale* **4**, 3065–3068 (2012); P. Xu et al, *Nat. Commun.* **5**, 3720 (2014)]. According to recent experimental works, [O. J. Wahab et al, *Nature* **620**, 782–786 (2023); Z. F. Wu et al, *Nat. Commun.* **14**, 7756 (2023)], these nanoripples reduce the energy barrier for proton transport in graphene, hBN and graphene oxide. We believe the same should

happen in monolayer titania. Following the Reviewer comment, we have incorporated these discussions in the revised SI (Page 4).

The single-titanium vacancies were introduced spontaneously during the preparation of 2D titania sheets in water. Their density was reported to be determined by the amount of titanium atoms being substituted by lithium atoms in their bulk layered precursors. In this context, is it possible to tune the density of single vacancies by, for example, tuning the composition of the layered precursors? If so, this would definitely add more promise for applications as sieving membranes, as compared to those porous membranes prepared by top-down methods, say, etching or bombardment, because the latter membranes are suffering from a wide pore size distribution.

The density of vacancies can indeed be tuned via the composition of titanate precursors. According to the previous literature, monolayers of titania with molecular formula of $\text{Ti}_{0.91}\text{O}_2^{0.36-}$, $\text{Ti}_{0.87}\text{O}_2^{0.52-}$, $\text{Ti}_3\text{O}_7^{2-}$, $\text{Ti}_4\text{O}_9^{2-}$, $\text{Ti}_5\text{O}_{11}^{2-}$ and $\text{Ti}_{0.825}\text{O}_{1.825}^{0.36-}$ have been exfoliated from layered crystals of different compositions, corresponding to a porosity from ~9% up to ~18% [L. Z. Wang et al, *Chem. Rev.* **114**, 9455-9486 (2014); T. Gao et al, *J. Mater. Chem.* **19**, 787-794 (2009)]. We have added this discussion in the main text (Page 8).

The authors reported careful proton transport measurements through an individual 2D titania crystal. The referee understands that from a fundamental viewpoint, the results demonstrating a high proton conductivity through the material and complete blockage of other ion and gas species are new and novel to the community. The presented experimental results are already sufficient for publication. Nonetheless, in terms of practical applications for, say, proton conductors, large-area membranes are needed. The authors are suggested to discuss potential ways of extending the results obtained from a single crystal to a larger scale, for example, any possible strategies to prepare large-area membranes for future research.

We thank the Reviewer for this positive assessment. We believe that the self-assembly techniques reported in the literature are promising routes towards applications. Relevant examples include layer-by-layer (LBL) assembly [Osada, M. et al. *Adv. Mater.* **18**, 1023–1027 (2006)] and Langmuir–Blodgett (LB) deposition [Matsuba, K. et al. *Sci. Adv.* **3**, e1700414 (2017)]. These methods enable fabricating high-quality and large-area titania membranes with a controlled number of layers. Importantly, the application of these self-assembled films could extend beyond proton conducting membranes. An area of opportunity widely explored in the literature is using 2D crystals as barrier layers for catalysts to enhance their activity [Hu, K., et al. *Nat. Commun.* **12**, 203 (2021); Kosmala, T. et al. *Nature Catalysis* **4**, 850-859 (2021)]. The implementation of this application should face less challenges than proton conducting membranes, as the assembly would take place directly on the target substrate. Following the Reviewer comment, this discussion has been incorporated in the main text (Page 2 and 8).

Reply to comments of Reviewer #2

In this work, the authors show that monolayer titania is highly permeable to protons while remaining impermeable to helium and small ions including Cl^- and Li^+ . There are still some questions needed to be elucidated before further consideration.

We thank the reviewer for all the helpful comments. We believe these have helped clarify several points in the manuscript.

1. How to control the number and distribution of titanium atom vacancies on each monolayer of titania. This irregularity may lead to differences in conductivity between different samples.

According to previous work [L. Z. Wang et al, *Chem. Rev.* **114**, 9455-9486 (2014); T. Gao et al, *J. Mater. Chem.* **19**, 787-794 (2009)], by tuning the molar ratio of ingredients in the mixture for preparing layered precursors, titania monolayers with molecular formula of $\text{Ti}_{0.91}\text{O}_2^{0.36-}$, $\text{Ti}_{0.87}\text{O}_2^{0.52-}$, $\text{Ti}_3\text{O}_7^{2-}$, $\text{Ti}_4\text{O}_9^{2-}$, $\text{Ti}_5\text{O}_{11}^{2-}$ and $\text{Ti}_{0.825}\text{O}_{1.825}^{0.36-}$ have been prepared, corresponding to a porosity ranging from ~9% up to ~18%. However, our experimental sensitivity in proton transport experiments ($2.0 \pm 0.8 \text{ S cm}^{-2}$) is about 40%, which is greater than the described range of porosity. Such variations are therefore not expected to result in a measurable difference in proton conductivity. Following the Reviewer comment, we have incorporated this information in the revised manuscript (Page 8) and discussed in SI (Page 2-3).

2. In HRTEM images, the size of titanium atom vacancies is about 0.1-0.2 nm, which does not correspond to the simulated vacancy size of 0.3-0.4 nm. It is recommended to supplement with single-crystal analysis.

In the schematic in Fig. 1a, the described size of $0.38 \text{ nm} \times 0.3 \text{ nm}$ corresponds to the unit cell parameters, which have been well-established by XRD and HRTEM [Sasaki, T., et al. *J. Am. Chem. Soc.* **118**, 8329–8335 (1996); Ohwada, M., et al, *Sci. Rep.* **3**, 2801 (2013); refs. 23, 26]. However, the space available for proton permeation is smaller than this interatomic separation, due to the electronic cloud surrounding the atomic nuclei. To clarify this point, we have included a DFT calculation of the electron density in the pristine material and in the presence of a Ti vacancy (Fig. S5a).

3. During the preparation process, an ion exchange process was used to replace lithium and potassium ions with protons to achieve delamination. This indicates that mobile cations and protons remain in the system, raising the question of how to exclude the influence of these cations. Due to the extremely low thickness of the titania monolayer, even trace amounts of cations and protons could cause significant deviations in the conductivity measurements.

The Reviewer suggests that the presence of cations (such as Li^+ and K^+) and protons from the ion exchange process may affect the conductivity measurements. This is a helpful comment. Such deviations could arise from cations blocking vacancies and hence decreasing the proton conductivity, or from residual protons yielding a higher surface proton concentration and hence higher conductivity.

To assess these possibilities, we note that the areal conductivity of devices measured with HCl electrolyte displayed a clear dependence on proton concentration. In contrast, for all tested salt solutions, including KCl, LiCl, MgCl_2 , CaCl_2 and $\text{Ru}(\text{bipy})_3\text{Cl}_2$, the devices displayed a finite conductivity σ_0 that was independent of the salt concentration (Fig. 4). This shows that the tested ions cannot permeate through the titania membranes – nor can they appreciably block the pores. Hence, the only possible contribution from the exchange process to the measured conductivity could be that protons from the exchange process remain adsorbed on the negatively charged titania surfaces. We agree that this is a possible explanation for the origin of σ_0 . However, note that this would only affect the conductivity of the membranes in the low proton concentration regime ($< 10^{-5} \text{ M}$), since for higher concentration, protons from the bulk electrolyte dominate the response. Following the Reviewer comment, we have amended the discussion on this possibility in the revised manuscript (Page 6-7). Thank you for this helpful comment.

4. Although the short-term thermal stability of titania has been confirmed, there is a lack of research on its performance under long-term high-temperature exposure. This is crucial for evaluating the material's

lifespan in practical applications. For example, titania has two main crystalline forms: anatase and rutile. Especially at high temperatures, anatase tends to convert to rutile.

Our manuscript is focused on the fundamental demonstration of proton transport through monolayer titania, which had not been investigated. We worked with microscale devices in a highly controlled device geometry (freestanding titania membranes suspended over a micrometer size aperture), which is suitable for the assessment of the material's fundamental properties. However, longer-term stability of the 2D membranes is difficult to assess. Our devices were usually found broken after a couple of days, which we attribute to the strain induced by variations in the temperature/humidity and by creep of the Nafion coating in our suspended devices. Such ruptures did not happen if we placed titania sheets on a silicon substrate, even at elevated temperatures (Fig. S6c). To demonstrate long term stability, we would need to design different devices targeted to the specific application in mind. However, this is not straightforward either. As we outline in our response to point #7, the applications of titania could extend beyond proton conducting membranes, to include, for example, coatings for catalysts. Accordingly, our manuscript is focused on the fundamental question of whether a titania monolayer is permeable to protons.

Regarding the crystalline phase of our samples, 2D titania crystals are in a lepidocrocite type [T. Sasaki et al, *Chem. Mater.* **10**, 4123-4128 (1998)] and phase transformation starts at $T > 300$ °C [H. Sutrisno et al, *Indo. J. Chem.* **10**, 143-148 (2010)]. In all our measurements, T was limited to <300 °C to avoid possible phase transformations. This is consistent with our XPS characterizations for samples before and after annealing at high temperatures (T up to 300 °C) in various gas atmospheres (Fig. S6a,b) that did not display any noticeable changes in their chemical structure.

5. Titania is a good electronic conductor. It is recommended to include ion migration number tests to distinguish between the contributions of ion and electron conductivity.

Previous direct characterization of the band structure of 2D titania showed that the material exhibits a large bandgap of about 3.8 eV, making it a good insulator [N. Sakai et al, *J. Am. Chem. Soc.* **126**, 5851-5858 (2004)]. Nevertheless, to quantify the possible contribution of electrons to the conductivity of our devices coated with porous Pt films, we measured them in either a hydrogen atmosphere or in vacuum. In the presence of H_2 , Pt absorbs H_2 and provides protons for transport through titania, whereas in a vacuum environment, only electrons contribute to the observed conductivity. As shown in Fig. 3, the observed conductivities in vacuum were three orders of magnitude lower than that in hydrogen in the full range of measurement temperature. This corroborates that the high conductivity observed in hydrogen was due to proton conduction and electron transport provided a negligible contribution to the overall conductivity, consistent with described titania's large bandgap.

6. The proton conductivity of pure water at room temperature is very low, about 5.5×10^{-5} $\mu S/cm$. The measured conductivity of KCl and LiCl solutions is about 10^{-2} S/cm, indicating that lithium and potassium ions can permeate through the titania monolayer.

We apologize for the confusion between 'areal conductivity' (S/cm^2) and 'bulk conductivity' (S/cm). Our I - V measurements through a 2D membrane yielded areal conductivities in unit of S/cm^2 . This confusion has now been clarified by changing the term 'conductivity' to 'areal conductivity' where necessary throughout the revised manuscript (see comment from Reviewer #4).

Following comment #3, the finite areal conductivity (about 10^{-2} S/cm^2) observed for DI water (Fig. 4c), which is insufficient to be accounted for by its bulk conductivity, can be attributed to a high density of

residual protons that adsorbed on the negatively charged titania surface. These protons could either come from water (a large electrolyte reservoir, even with high bulk resistivity, does not limit the conductance of a very small membrane), or as the Reviewer suggested, from the proton exchange step during preparation of the layered precursors. The fact that similar areal conductivities ($\sim 10^{-2}$ S/cm²) were detected for all tested salt solutions, which were independent of the solution concentration and ion types (Figs. 4c,d), unambiguously corroborated that the bulky ions including Li⁺ and K⁺ were impermeable.

7. Given that many two-dimensional nanosheets have been proven to conduct protons, it is expected that titania monolayers containing angstrom-scale titanium atom vacancies can be used to conduct protons. It is necessary to clarify the novelty of the paper and provide practical application tests to demonstrate its potential.

While one could have reasonably expected titania to be proton permeable due to its high density of angstrom-scale pores, it was unexpected that the material's properties would be so good. It is highly permeable to protons while remaining impermeable to helium and small ions including Cl⁻ and Li⁺; its proton conductivity exceeds the targets set by industry roadmaps; it operates at $T \approx 200$ °C; and it is a widely accessible material. Besides those sought-after properties, another area of opportunity widely explored in the literature is using 2D crystals as barrier layers for catalysts to enhance their activity [Hu, K., et al. *Nat. Commun.* **12**, 203 (2021); Kosmala, T. et al. *Nature Catalysis* **4**, 850-859 (2021)]. A more recently discovered possibility is exploiting these materials' atomic thinness to accelerate electrochemical processes via electric field effects [Tong, J., et al. *Nature* **630**, 619–624 (2024)]. Following the Reviewer comment, we have expanded the introduction to emphasize the novelty of the work and include some of the possibilities in the revised manuscript (Page 2).

To implement practical application tests, large-area membranes supported on suitable substrates are needed. This is conceptually different from our devices, which were designed to demonstrate the point that a 2D material with a high density of vacancies can be an effective alternative to pristine 2D crystals. To complicate matters, the possible applications are not restricted to proton conducting membranes, as mentioned above, so more than a single proof of concept device would be required. Given these limitations, we cannot perform practical application tests at this stage. The discussion of such tests would overload our manuscript, which already contains a huge amount of experimental effort. These new experiments merit a dedicated study. Nonetheless, we point out that large-area titania membranes have been reported in recent years via self-assembly techniques such as layer-by-layer (LBL) assembly [Osada, M. et al. *Adv. Mater.* **18**, 1023–1027 (2006)] and Langmuir–Blodgett (LB) deposition [Matsuba, K. et al. *Sci. Adv.* **3**, e1700414 (2017)].

8. Two-dimensional crystal materials such as graphene and hexagonal boron nitride (hBN) can effectively sieve hydrogen isotopes. These one-atom-thick crystals are impermeable to atoms and molecules but allow hydrogen ions (thermal protons) to penetrate. How to prove that the proton conduction mechanism of titania monolayers differs from that of graphene and hBN? Does proton transport occur through titanium atom vacancies?

In the absence of vacancies, titania has a dense electronic cloud that poses an unsurmountable energy barrier to protons of over 1 eV (see response to point #2 and Fig. S5). The material becomes proton permeable only through the introduction of Ti vacancies, which effectively form proton conducting channels through the lattice (see DFT calculations in Fig. S5 and comments from Reviewer #3). This is a different permeation mechanism to that observed in graphene and hBN in which the one-atom-thick

structure is intrinsically proton permeable – albeit with much lower conductance than titania. Our isotope effect measurements (Fig. 4c) also demonstrate that the permeation mechanism is different. These yielded $\sigma_{\text{H}}/\sigma_{\text{D}} = 1.6 \pm 0.16$ for titania, which is an order of magnitude smaller than the $\sigma_{\text{H}}/\sigma_{\text{D}} \approx 10$ observed for graphene and hBN monolayers [M. Lozada-Hidalgo et al, *Science* **351**, 68-70 (2016), ref. 2].

9. Generally, the proton conductivity of materials is affected by spatial hindrance. The pore size of the titania monolayer is less than 1 Å, so obtaining proton conductivity likely requires very high temperatures, making it difficult to detect proton conductivity at low temperatures.

We fully agree. As the Reviewer commented, normally proton transport through such small pores requires high temperatures to yield a measurable current. However, while the pores in our films are angstrom-scale, their length is in the nanometer scale. Proton transport at room temperature in pores with such dimensions is well established, for example carbon nanotubes or gramicidin (see comment #4 from Reviewer #3), which is consistent with our findings.

Reply to comments of Reviewer #3

This work reports high proton conductivity through angstrom-porous titania membranes. The authors prepared 2D titania, i.e., 2D array of TiO₆ octahedra, and utilized the high density of angstrom-scale Ti-atom vacancies for proton transport across the 2-D membrane. After carefully characterizing the titania membrane, the authors first measured proton transport at low temperatures (i.e., from room temperatures to 60 °C) using a Nafion-based device where the 2-D membrane is coated with Nafion and proton-injecting electrodes (Pt on carbon) are used for electrical connections. They measured a proton conductivity of 2 S/cm² for the titania membrane at room temperature which is one or two orders of magnitude higher than that for other 2-D membranes and found the activation energy for proton transport is 0.34 eV. The authors then measured the membrane proton conductivity at temperatures ranging from 20 to 260 °C using a Pt-coated device where porous Pt films are coated on both sides of the titania membranes as the proton source and the electrodes. They found that the membrane proton conductivity reaches 200 S/cm² at 260 °C and the activation energy measured from the Pt-coated device is very similar to that measured from the Nafion-based device. Finally the authors used an aqueous system where two sides of the membrane were filled with aqueous solutions and Ag/AgCl electrodes were used for electrical connections to demonstrate the excellent proton selectivity over other ions.

Overall, this work is a complete and excellent work. The high proton conductivity of the 2D titania membrane could really make it a good proton conductive material within the well-known proton materials gap, holding great promise for a variety of hydrogen-based technologies. The paper is well-written and the proton transport measurements using the three different systems were nicely designed and performed. Personally, I think this work meets the requirements of Nature Communications. However, there are still some problems related to the claims and the explanations. I would like to ask the authors to address and/or clarify those issues before recommending its publication in Nature Communications. Please find my detailed comments below.

We are very grateful to the Reviewer for this positive assessment of our work. We have incorporated all the Reviewer's comments, which have improved the clarity of our manuscript and have allowed us to link our findings to the rich nanopore literature.

1. The proton conductivity of the titania membrane measured from the Nafion device is quite different from that measured from the Pt coated device. The authors attributed that to possible Pt blockage of the vacancies or the lower proton density on the surface of titania for the Pt coated device. For the second possible explanation, is it possible to estimate the proton density inside the porous Pt thin film at the experimental conditions? It would be great if the authors can provide such an estimation and compare it with the proton concentration (~0.1M) inside the Nafion film.

The proton conductivity of devices measured with Nafion and Pt coating layers differ by a factor of ~3. This is larger than our experimental accuracy (typically a factor of ~2 between different devices), but it is small compared to the orders-of-magnitude difference in conductivity observed in other 2D crystals (e.g. hBN and graphene, Fig. 2b). Estimating the proton concentration in Pt would indeed be helpful. However, to explain the relatively small difference in conductivity between the Pt- and Nafion-coated devices, which is just above our experimental accuracy, we would need precise estimates of the active surface area and the density of protons on the porous Pt layer. We are not confident that we can provide these estimates with the accuracy required for the evaporated thin films.

*2. The analysis and discussion for the drift-diffusion experimental result is not correct. The total diffusion voltage measured at zero current should include contributions both from the selective transport and the electrode redox reaction. $V = V_{transport} + V_{electrode} = -(t_H - t_{Cl}) * k_B * T / e * \ln(\Delta C) - k_B * T / e * \ln(\Delta C)$ according to the authors' measurement configuration. However, the authors totally ignored the second term in their analysis and discussion. As the second term is -59 eV for a Cl⁻ concentration gradient of 10 across the membrane, the first term, $V_{transport}$ is actually close to zero, indicating that the titania membrane has no ion selectivity at all for concentrations above 0.1 M. I do not understand this result since the conductivities of 1 M KCl and LiCl are four orders of magnitude lower than the conductivity of 1M HCl and thus the membrane seems to only allow proton transport even for 1M solutions. The authors should explain this conflict between the drift-diffusion result and the ion selectivity result. Also, I would suggest that the authors reconduct the drift-diffusion experiments but change the HCl concentrations on two sides of the membranes to 0.1 M and 10 mM or to 10 mM and 1 mM respectively. They may be able to use these new results to better show the ion selectivity if surface-charge-governed ion transport play an important role on the ion selectivity.*

We apologize for not making our methods clear in the text. In the manuscript, we removed the contribution from the electrodes to the cell voltage, following the methodology in our previous work [L. Mogg, et al *Nat. Commun.* **10**, 4243 (2019); ref. 3]. This allows focusing the discussion on the membrane potential, which is the central point of that experiment. We find that removing this technical element makes our results more accessible to a wider audience. However, we acknowledge that this was not made clear enough in the text, which can lead to confusion for expert readers. Following the Reviewer comment, we have clarified this point on Page 6 of the main text. Additionally, the x-axis on Fig. 4b is now labelled 'membrane potential' and we have expanded the SI to show our raw data (Fig. S8), which includes the potential drop at the electrodes.

3. The authors claimed that the activation energy measured from the Nafion device and the Pt devices is almost the same and thus the transport mechanism is the same for both type of devices. However, the Arrhenius plots for the Pt-coated devices (Fig. 3 and Fig. S7) are not that linear. In particular, the conductivities measured at higher temperatures deviate from the linear fitting, indicating possibly a larger

activation energy for higher temperatures. The authors should try to explain why there is a nonlinear temperature dependence and why the activation energy would increase at higher temperatures.

The Arrhenius plots for Pt-coated devices deviate slightly from the exponential fit at elevated temperatures. It is difficult to explain this small feature conclusively. However, we suggest that it could be related to the properties of adsorbed water on the surfaces of our devices. Such water layer can be expected to facilitate proton injection from the Pt film to the titania surface, similar to the so-called 'triple phase boundary' between catalyst, hydrogen gas and proton membrane in fuel cells. The equilibrium concentration of this adsorbed water layer in the humid environment in the experiments can be expected to decline at higher temperatures, which could explain the increase in the activation energy of the proton transport process. Following the Reviewer's comment, we have briefly discussed this possibility in the Supplementary Information Section 'High-temperature measurements' (Page 5).

4. While the authors performed a thorough experimental study to show fast and selective proton transport across the titania membrane, they only provide a very short discussion/explanation about the corresponding mechanism at the end of the manuscript. I would suggest that the authors expand the mechanism discussion in the revised manuscript. Specifically, How large is the measured activation energy when compared with those measured in other small channels or pores? (e.g., J Shao et al. Nature Communications 2015 and R. Tunuguntla et al., Nature Nanotechnology, 2016)? How do protons transport across the membrane? Is it based on the Grotthuss mechanism or regular vehicle mechanism? The authors should consider moving part of the DFT result discussion in the supplementary information to the main text to better explain the mechanism. Also, it would be interesting to compare the proton conductance of a single titanium vacancy with those from natural or artificial small channels/pores, e.g., gramicidin A channels and sub-1 nm carbon nanotube porins. Would the 2D titania vacancy exhibit much higher proton conductance?

Our understanding of the transport mechanism is that the oxygen-containing pores have a large negative charge, as evidenced by our zeta potential measurements (Fig. S9), which results in high proton concentration in the crystals. When a voltage bias is applied across the devices, protons from surrounding media (electrolyte or Pt films) are injected into the crystal (see Point #3), leading to a proton current. Our DFT calculations provide insights into the mechanism for transport inside the crystals. These show that protons can hop between oxygen groups in the pores, similar to our previous work in atomically thin micas [Mogg, L. et al. *Nat. Nanotechnol.* **14**, 962-966 (2019); ref. 12]. This mechanism resembles proton hopping along water chains in the Grotthuss mechanism, except water molecules are replaced by oxygen groups. The calculated energy barrier for this hopping mechanism (≈ 0.4 eV) agrees well with the one extracted experimentally (≈ 0.36 eV), validating the DFT calculation. Following the Reviewer comment, we have included this discussion in the main text of the revised manuscript (Page 8).

It is a helpful suggestion from the Reviewer to compare the conductivity per pore in our titania membranes with the rich nanopore literature. From the concentration of pores in our crystals ($\sim 10^{14}$ cm $^{-2}$) and the conductivity of the devices (≈ 2 S cm $^{-2}$), we estimate that the conductance per pore is $\sim 10^{-6}$ - 10^{-5} nS, with the uncertainty arising from the accuracy in estimating pore density. One key reference system is the work of R. Tunuguntla et al. *Nature Nano.* 2016. The authors report carbon nanotubes 0.8 nm in diameter and 10 nm in length. Normalising by length (multiplying by a factor of 10), these pores displayed proton conductance of $\approx 3.3 \times 10^{-6}$ nS. The corresponding activation energy was 0.57 eV. Another reference is the biological channel gramA, which is 0.4 nm in diameter and 5 nm long. The length-normalised conductance is $\approx 0.6 \times 10^{-6}$ nS and the activation energy is ≈ 0.2 eV. Although there are differences in the

conductance and energy barrier of all these pores, the most striking finding from this comparison is how similar they all are. All the conductance values are roughly around $\sim 10^{-6}$ nS within a factor of about 10 and the energy barriers are around 0.3 eV within a factor of about 2. Following the Reviewer comment, we have included this discussion in a new Section ‘Comparison with other proton-conducting pores’ of the revised SI (Page 6-7).

5. When discussing the proton conductivity at elevated temperatures, the authors compared the conductance of an angstrom thick titania membrane with a 200- μm -thick Nafion membrane. This comparison seems unfair because of the significant difference in membrane thickness. Thinner Nafion membranes would definitely exhibit much higher conductance. I believe that the reason why industry choose the 200- μm -thick Nafion membrane is more from a durability perspective (including both mechanical and chemical stability). Will the 2-D titania membrane provide similar durability as the thick Nafion membrane? I would suggest that the authors add a short discussion on this to better justify the titania membrane will have promising applications for hydrogen-based technologies. Also, the authors should do a better job in justifying why we need 2D membranes as proton conducting membrane materials for hydrogen-based technologies at the beginning of the manuscript. Is it mainly because of the proton materials gap? Nevertheless, the mechanical stability of 2-D membranes will always be an issue for practical applications.

Thick Nafion membranes are necessary to prevent water and hydrogen crossover in fuel cells and electrolysers. To achieve the complete impermeability to water and hydrogen we observe in our titania films, 200 μm or even thicker Nafion films would be necessary. The appeal of titania is that complete crossover suppression can be achieved with just a 1 nm film. That is one of the justifications to use 2D crystals in hydrogen technologies: perfectly selective proton permeable barriers. Another justification, particularly in the case of titania, is that few materials can operate effectively around 200 – 300 °C. Our introduction focused on this point because this is a key competitive advantage of 2D titania (perfect selectivity can be achieved by, say graphene, albeit with lower proton conductance). Another area of opportunity widely explored in the literature is using 2D crystals as barrier layers for catalysts to enhance their activity [Hu, K., et al. *Nat. Commun.* **12**, 203 (2021); Kosmala, T. et al. *Nature Catalysis* **4**, 850-859 (2021)]. The implementation of this latter application should face less challenges than proton conducting membranes, as the films would be supported directly on the target substrate. Another, more recently discovered possibility, is exploiting these materials’ atomic thinness to accelerate electrochemical processes via electric field effects [Tong, J., et al. *Nature* **630**, 619–624 (2024)]. Following the Reviewer’s comment, we have expanded the introduction to include some of this discussion (Page 2).

6. Fig.4c clearly shows that the proton concentration inside the 2-D membrane depends on the surrounding medium concentration and there is a surface-charge-governed proton transport regime. It is great to see that the authors did zeta potential measurement for the titania surface and estimate the surface charge density. I would also suggest that the authors estimate the corresponding proton concentration inside the membrane below which surface charges dominate the proton transport using the expression $c = 4 \cdot \sigma / \text{diameter} / e / 1000 / N_A$. They should check if this estimation can explain the observed HCl concentration dependence observed in Fig. 4c.

This is indeed our understanding of the obtained results. The transport characteristics display a clear surface-charge governed regime which arises from the high charge density in the membranes. The Reviewer’s suggestion to estimate the proton concentration inside the pore volume is interesting.

However, we are not confident that this would be an accurate representation of the system. One problem is that our DFT calculations show that the excess charge due to the Ti vacancies is not fully concentrated inside the pores. As a result, some of the adsorbed protons would be on the surface of the crystal, away from the pores. Hence, proton diffusion from the crystal surface into the pores can be expected to contribute significantly to the overall conductance. Another problem is the definition of proton concentration in M L^{-1} under such extreme confinement. As discussed in Point #4, we do not expect water molecules to be present inside the pores. Hence, while it is possible to define concentration based on the pore volume, this estimate would not be necessarily comparable to the proton concentration in the aqueous electrolyte. Despite these limitations, our zeta potential measurements do provide insights into the surface-charge-dominated regime. We find that for large pH the zeta potential is effectively constant but it starts shifting for $\text{pH} < 3$, similar to the dependence observed in our transport measurements. A more quantitative analysis is not possible, because the estimation of surface charge from the zeta potential requires small potentials, which are only achieved in our system for $\text{pH} \approx 1$.

7. There are no error bars for the conductivity data in Fig. 2c. Also, personally, I do not like the grey “representative” symbols with the error bars in Fig.3 and Fig. 4d. Since the measured conductivity varies from 0.01 to 100 S/cm², how could a single symbol in each plot represent all error bars? I would suggest that the authors add error bar for each data point in those two figures.

Thanks for the comments and suggestions. All our T dependence data shown in the same figure were collected from the same device with the error bars (shown only if larger than the symbols) indicating standard deviations (SD) from multiple independent measurements. Specially, in Fig. 3, SD using different devices (for example, the ones in Figs. 3 and S7) increased with increasing T but did not exceed 50% of the measured values for all temperatures; in Fig. 4d, SD were less than 40% of all the solutions. We apologize for the confusion caused to readers. Following the Reviewer’s suggestion, and also to keep the figures as clear and simple as possible, we have removed the grey symbols with error bars from Figs. 3 and 4d and instead, specified details about SD in the figure captions.

Reply to comments of Reviewer #4

The authors present a study of ion and gas transport through freestanding titania monolayers supporting atop micrometer-sized holes in silicon nitride supports, with each side of the layers accessible to varying solutions and/or gases. They find that titania layers free of any large mechanical defects are excellent barriers to all species except proton over a very wide temperature range, from ambient conditions where protons exist principally as hydronium ions, to >200 C where protons lack solvation. Areal conductances are quite high for protons and are orders of magnitude lower for other ions.

The work extends prior work from the same group on proton conduction through other 2D materials, notably graphene and boron nitride. The work is well executed with many control experiments to rule out potential artefacts from macroscopic defects. The lack of any gas transport in He leak tests, the clear and repeatable difference between apertures with and without titania layers, the near-zero conductance in the absence of hydrogen, and the expected trends in cell potential for solutions of differing HCl concentration on the two sides of the membrane, all are consistent with their interpretation of high proton conductance and high proton transference through titania layers. Their statements regarding the possible role of proton attachment to broken bonds of Ti-atom vacancies is sensible and consistent with the similarity of

mechanisms over the wide temperature range which includes conditions where protons are primarily solvated, and where they are not solvated.

We thank the reviewer for such a positive assessment of our work. We have revised the manuscript accordingly following the two comments.

I have only two minor comments. First, the manuscript is inconsistent in usage of the term “conductivity”, with the term in some places being correctly identified as areal conductivity (i.e., conductance normalized by area), and in some places the simple term “conductivity” is used. Conductivity is a bulk property and has units of $S\text{ cm}^{-1}$; that is not how the term is used here. I recommend that in all instances where the term “conductivity” is used, that the term is modified to always state that it really means “areal conductivity”. In this way, confusion with other studies that do truly report bulk conductivity will be avoided.

We have implemented this suggestion and changed the term “conductivity” to “areal conductivity” where necessary in the revised manuscript.

The second comment is more general and has to do with the possible effects of hydrogen gas on titania properties. Titania normally has a Ti valence of 4+ which makes it a wide-bandgap insulator, but contact with hydrogen gas can reduce some Ti atoms to the 3+ valence state, which then makes the material mixed-valent and a potential electronic conductor. These changes could be highly susceptible to environmental exposure and could result in a situation where the titania layers become doped on exposure to hydrogen, and then de-doped as hydrogen is removed. I'm not sure what the best test for this would be, but I present the possibility as something for the authors to consider.

We are grateful for this insightful comment. Let us first note that the T dependence of the observed areal conductivities was practically the same in Nafion, vacuum and hydrogen, showing the activation energy of ~ 0.4 eV. If hydrogen would dope titania, leading to electron rather than proton transport, one should probably expect changes in the activation energy as different mechanisms usually result in different energy barriers. This allowed us to conclude that we dealt with proton transport in all the cases. Nonetheless, following the Reviewer comment, we have carried out two additional experiments to check this conclusion. 1) We performed XPS characterization of the titania sheets after annealing in a hydrogen atmosphere (300 °C for several hours). No discernible changes in the peaks assigned to Ti $2p_{1/2}$ and Ti $2p_{3/2}$ bands could be observed with respect to those before annealing and, also, after annealing in either air or Ar (new Fig. S6). 2) We also measured in-plane conductivity of titania monolayers using micron-spaced electrical contacts placed on the surface of individual flakes. No difference in conductance was found between vacuum and hydrogen cases even at the highest applied temperature of 250 °C (only leakage currents were detected as expected for titania being an insulator with the large gap of 3.8 eV).

RESPONSE TO REVIEWERS' COMMENTS

Reply to comments of Reviewer #1

The authors have addressed all concerns and the manuscript has been improved significantly. Now I can recommend its publication.

We thank the Reviewer for the recommendation.

Reply to comments of Reviewer #2

Most issues have been addressed. It can be accepted.

We thank the Reviewer for the support with acceptance of our revised paper.

Reply to comments of Reviewer #3

The authors have done an excellent job in addressing my comments in the revised manuscript. I therefore recommend publishing this paper in Nature Communications.

We are very grateful to the Reviewer for this recommendation.

Reply to comments of Reviewer #4

The authors have adequately addressed my comments from earlier review, particularly involving issues associated with mixed valence in partially reduced titania.

We thank the reviewer for this positive assessment and useful comments assisting our revision process.